# In-Situ Surface Modification of ITO Substrate via Bio-Inspired Mussel Chemistry for Organic Memory Devices

**DOI:** 10.3390/biomimetics7040237

**Published:** 2022-12-12

**Authors:** Minglei Gong, Wei Li, Fei Fan, Yu Chen, Bin Zhang

**Affiliations:** 1Key Laboratory for Advanced Materials and Joint International Research Laboratory of Precision Chemistry and Molecular Engineering, School of Chemistry and Molecular Engineering, East China University of Science and Technology, Shanghai 200237, China; 2Shanghai i-Reader Biotech Co., Ltd., Shanghai 201100, China

**Keywords:** memory device, mussel chemistry, surface modification, SI-ATRP

## Abstract

The development of organic memory devices, regarding factors such as structure construction, principle exploration, and material design, has become a powerful supplement to traditional silicon-based information storage. The in-situ growth of materials on substrate surfaces can achieve closer bonding between materials and electrodes. Bio-inspired by mussel chemistry, polydopamine (PDA) was self-assembled on a flexible substrate as a connecting layer, and 2-bromoiso-butyryl bromide (BiBB) was utilized as an initiator for the polymerization of an iridium complex via surface-initiated atom-transfer radical polymerization (SI-ATRP). A device with the structure of Al/PDA-PPy_3_Ir/ITO was constructed after the deposition of aluminum. The device exhibited a nonvolatile rewritable memory characteristic with a turn-on voltage of −1.0 V and an ON/OFF current ratio of 6.3 × 10^3^. In addition, the memory performance of the Al/PDA-PPy_3_Ir/ITO device remained stable at bending states due to the intrinsic flexibility of the active layer, which can be expanded into the establishment of flexible memory devices. Spectroscopy and electrochemical characterization suggested that the resistive memory properties of the device stemmed from charge transfer between PDA and iridium polymer in the active layer (PDA-PPy_3_Ir) under an applied voltage.

## 1. Introduction

The storage of ever-increasing amounts of information is an inevitable problem due to rapid advancements in science and technology [1]. When the amount of data to be stored expands to the zettabyte magnitude, higher requirements will be placed on memory devices [2]. So far, numerous novel devices based on new mechanisms, materials, and structures have been developed to respond to the limits of Moore’s law and solve the electric leakage problem in traditional silicon-based memory devices [3,4,5,6]. Among them, organic resistive memory devices have become highly anticipated due to their low costs, ease of processing, excellent scalability, and flexibility [7,8,9]. The active layers of organic memory devices range from simple small molecules [10], polymers [11], complexes with various ligands and coordination centers [12,13], and other biomolecules [14]. Metal complexes, especially transition metal complexes, show great potential in resistive memory devices due to the range of options of central metals and ligands with different functions [15]. For instance, Goswami and co-workers [16] found that rhodium coordinated with three azopyridine ligands that lacked memory properties and exhibited non-volatile memory behavior with a large ON/OFF current ratio of about 103. Similar results were obtained when they changed the metal to iridium [17]. Liu and co-workers [18] synthesized three kinds of conjugated polymeric materials containing cationic Ir(III) complexes in the main chain by changing the ligand structure of the iridium complex unit. Ir(III) complexes served as electron acceptors (A) in the polymer resistive memory device.

PDA is a bionic inspired by the adhesive mussel byssus protein. Under alkaline conditions, dopamine self-polymerizes, even on smooth surfaces including metals, glass, ceramics, and plastics [19]. Compared with traditional spin-coated polymer-based memory devices, the in-situ growth of PDA on a substrate can achieve easier and closer bonding between materials and electrodes due to its simple preparation method and excellent adhesiveness [20,21,22]. Moreover, the surface of PDA is rich in catechols and amines, which can act as reaction sites with small molecules, polymers, or metal complexes [23]. By forming such a multifunctional reaction platform, the applications of PDA have been further expanded to chemical modification, and studies of PDA in resistive memory devices have also recently emerged [24,25]. It is worth mentioning that PDA also shows great potential applications in flexible and wearable memory devices [26].

In this contribution, PDA was self-assembled on an indium tin oxide (ITO)-coated polyethylene glycol terephthalate (PET) substrate as a reaction platform. By adding the bromine-containing initiator, BiBB, surface-initiated atom-transfer radical polymerization (SI-ATRP) was triggered to graft an iridium complex on PDA to form a polymeric active layer. The characterization of the structure and morphology confirmed that PDA and iridium polymers were in-situ and grown step-by-step on the substrate. The thin film formed by PDA and iridium polymers had a planar surface morphology with a thickness of about 80 nm. Then, a flexible device with the structure of Al/PDA-PPy_3_Ir/ITO was fabricated by evaporating ultra-pure aluminum as the top electrode. The device exhibited nonvolatile rewritable memory behaviors with a turn-on voltage of −1.0 V and an ON/OFF current ratio of 6.3 × 10^3^. The memory behaviors of the flexible device were almost unchanged under the bending test. To compare SI-ATRP with the traditional spin-coating process, a heterojunction device Al/PPy_3_Ir/PDA/ITO was constructed by self-assembly of PDA on ITO/PET substrate and spin-coating of the polymerized iridium complex on the PDA layer. The results demonstrated that, though the contrast device also exhibited nonvolatile rewritable memory characteristics, the Al/PDA-PPy_3_Ir/ITO device prepared by SI-ATRP had a higher current switching ratio and reliability.

## 2. Materials and Methods

### 2.1. Materials

IrCl_3_·3H_2_O, 2-phenylpyridine, 4-(2-pyridine)-benzaldehyde, 2-ethoxy ethanol, methanol, silver trifluoroacetate, anhydrous sodium sulfate, sodium borohydride, ammonium chloride, chloroform, 4-phenylisulfonic acid, methacrylic acid, 2-bromoiso-butyryl bromide, and azo bisisobutyronitrile were analytically pure, purchased from Shanghai Aladdin Biochemical Technology Co., Ltd. (Shanghai, China), dried before use, and the synthesis procedure was carried out in anhydrous and argon atmosphere. PET substrate was coated with ITO (200 nm) on the surface, purchased from South China Xiangcheng Technology Co., Ltd. (Shenzhen, China), square resistance ≤ 6 Ω, cut into 3 × 3 cm squares before use.

### 2.2. Measurements and Instrument

^1^HNMR was measured with a Bruker 400 MHz (Bruker, Billerica, MA, USA)/AVANCE III 400 spectrometer (AVANCE, Hong Kong, China). ESCALAB 250Xi was used for XPS testing with Al Kα as the X-ray source. UV-Vis absorption spectra were measured with a Shimadzu UV-2540 spectrophotometer (Shimadzu, Kyoto, Japan). FT-IR spectra were measured with a Spectrum 100 infrared spectrometer. PL spectra were measured by HORIBA FluoroMax-4 spectrophotometer (Horib, Kyoto, Japan). FESEM images were obtained with a Nova Nano SEM 450 microscope. AFM images were obtained with a Solver P47-PRO (NT-MDT Co., Ltd., Moscow, Russia) microscope; The cyclic voltammetry and AC impedance curves were measured by CHI-650D electrochemical workstation. The electrolyte solution was 0.1 mol/L of an acetonitrile solution of tetrabutylammonium hexa-fluorophosphate. A three electrode system was operated: Pt (working electrode), Ag/AgCl (reference electrode), and platinum wire (counter electrode). The working electrode was placed in a stirring reactor to grow PDA-PPy_3_Ir film on the surface. EPR was measured with a Bruck EMX-8 paramagnetic resonance spectrometer. UPS was measured by ESCALAB 250 system. Raman spectra were measured with a Invia/Reflex Laser Micro-Raman spectrometer with an excitation wavelength of 532 nm and an argon ion laser as the light source.

### 2.3. Synthesis of Iridium Complex

The synthesis route of the iridium complex is illustrated in Appendix A. An amount of 150 mL of 2-Ethoxyethanol (150 mL) and 50 mL deionized water were added to a 500 mL round-bottom flask containing IrCl_3_·3H_2_O (1.41 g, 4 mmol) and 2-phenylpyridine (1.55 g, 10 mmol). The reaction mixture was refluxed for 24 h and then cooled to room temperature. Deionized water (150 mL) was added to the flask before filtration. A yellow chloride-bridged dimer iridium complex **1** (0.7 g, 0.65 mmol) was obtained by washing the filter cake with deionized water and methanol. The dimer iridium complex (0.65 mmol), silver trifluoroacetate (1.3 mmol), and 4-(2-pyridyl) benzaldehyde (1.5 mmol) were added to a round-bottom flask, followed by 50 mL of 2-ethoxyethanol, and refluxed for 48 h and then cooled to room temperature. Then, the residue was diluted with ethyl acetate, washed with water three times, and dried with anhydrous sodium sulfate. The product was concentrated by rotary evaporation, and a bright yellow product **2** (0.55 g, 62%) was obtained by silica gel column chromatography (DCM:EA = 5:1). Product **2** (0.6 mmol) was dissolved in 10 mL anhydrous ethanol, and 55 mg NaBH_4_ was added under an argon atmosphere and stirred at room temperature for 3 h. The reaction was quenched with 5 mL of saturated ammonium chloride solution. Ethanol was removed by rotary evaporation, the product was then extracted with chloroform three times (3 × 10 mL) and dried with anhydrous sodium sulfate. A yellow product **3** (0.3 g, 73%) was concentrated and mixed with methacrylic acid (0.9 mmol) and the catalyst, 4-phenylisulfonic acid. The mixture was heated at 150 °C for 12 h and cooled to room temperature before extraction. The organic phase was washed and concentrated, and final product **4** (0.18 g, 55%) was obtained by silica gel column chromatography (DCM:EA = 10:1). ^1^H NMR (CDCl_3_, 400 MHz)δ(ppm): 7.87–7.85(d,3H), 7.66–7.62(m,3H), 7.59–7.55(m,3H), 7.52–7.49(m,3H), 6.88–6.85(m,6H), 6.82–6.80(m,3H), 6.00(s,1H), 5.49(s,1H), 5.00(s,2H), 1.88(s,3H).

### 2.4. Preparation of Al/PDA-PPy_3_Ir/ITO Device 

A PET film coated with an ITO layer was used as the substrate and cut to the size of 3 × 3 cm before being successively cleaned with ethanol, acetone, and isopropyl alcohol for later use. Dopamine hydrochloride (50 mg) and 50 mL of 50 mmol/L Tris-HCl buffer solution (pH 8.5) were added to the reactor with stirring. After 5 min of ultrasonication, dopamine was dispersed, and the PET substrate was placed vertically in the reaction container. After 8 h, the substrate was taken out, and its surface was carefully cleaned with water and ethanol, successively. Then, it was dried overnight at 50 °C in a vacuum oven to obtain the PDA/ITO substrate. The active layer film was formed by SI-ATRP and dried overnight under vacuum to obtain the PDA-PPy_3_Ir/ITO substrate. Ultrapure aluminum was used as the top electrode and deposited on the surface of the active layer by electron beam evaporation to obtain an Al/PDA-PPy_3_Ir/ITO device with a sandwich structure. The vacuum degree of the evaporation process was 10^−7^ Torr. The radius of the electrode was 0.2 mm, and the thickness was 200 nm.

### 2.5. Surface-Initiated Atom-Transfer Radical Polymerization

The PET substrate was placed in a reactor containing 10 mL of ultra-dry dichloromethane and 3 mL trimethylamine. Initiator dibromoiso-butyryl bromide (1 mL) was diluted with 2 mL dichloromethane and dropped into the reactor under an argon atmosphere. The reaction mixture was stirred in a water bath at 30 °C for 24 h. After removal, the substrate was rinsed with water and ethanol, and dried under vacuum. Iridium complex **4** (30 mg) and 7.5 mg CuI were added to the reactor and dissolved in 15 mL of ultra-dry methylene chloride. The substrate was placed in the reactor under an argon atmosphere, and 50 μL PMDETA was added and then stirred for 30 h at 30 °C. After removal, the active layer PDA-PPy_3_Ir was successively washed with water and ethanol and dried under vacuum.

## 3. Results and Discussion

The preparation of the Al/PDA-PPy_3_Ir/ITO device is briefly illustrated in Figure 1. First, PDA was self-assembled as a connecting layer on the surface of an ITO-coated PET substrate. Then, the iridium complex (synthesis route shown in Appendix A) was polymerized on the surface of PDA via SI-ATRP with BiBB as the initiator [27,28,29,30]. Then, aluminum was evaporated on the surface of the iridium polymer under vacuum to obtain a device with a structure of Al/PDA-PPy_3_Ir/ITO. The detailed preparation procedure and conditions are shown in the Appendix A.

To confirm the growth of the active layer on the substrate surface, Fourier-transform infrared (FT-IR) spectroscopy, Raman spectroscopy, and X-ray photoelectron spectroscopy (XPS) were employed at different stages during the preparation of the active layer. By contrasting the position of new peaks in the XPS wide-scan spectra in Figure 1a, clear C 1s, N 1s, and O 1s peaks (green) were observed at 284.6 eV, 400 eV, and 529 eV, respectively, when PDA self-assembled on the surface to obtain the PDA/ITO substrate. In the C 1s core-level spectrum of this substrate in Figure 1b, the peaks at 284.6 eV, 285.5 eV, 286.2 eV, 287.5 eV, and 288.8 eV were attributed to C-H, C-N, C-O, C=O, and O=C-O, respectively [31], which suggested the successful growth of PDA. After anchoring the initiator, new peaks appeared in the wide-scan spectrum (blue). The peak in the core-level spectrum (Figure 1c) at 69.6 eV, which belonged to Br 3d derived from the terminal bromine atom of BiBB, proved the anchoring of the initiator [32]. Similarly, after the polymerization of the iridium complex (red), new peaks emerged at 60.3 eV and 63.3 eV in Figure 1d, which belonged to the Ir 4f_7/2_ and Ir 4f_5/2_ orbitals [33]. This implied the grafting of the iridium polymer during the preparation of the active layer. The characteristic peaks of methyl, amide bonds, and ester groups at 1385, 1612, and 1721 cm^−1^, corresponding to acyl bromide and iridium ligand, were observed in the FT-IR spectra in Figure 1e. These peaks indicated the introduction of BiBB and the iridium complex to the system [34]. As a supplement, Raman spectroscopy (Figure 1f) showed that the PDA-PPy_3_Ir/ITO substrate contained characteristic peaks at 1350 cm^−1^ and 1580 cm^−1^ for PDA and 1250 cm^−1^, 1440 cm^−1^ for the iridium polymer [35]. These spectroscopic characterizations provide strong evidence that active layer materials were in-situ modified on the substrate surface.

PDA inevitably aggregates in an alkaline solution. Small particles will deposit at the bottom and continuously accumulate, which affects the surface morphology. To obtain a uniform surface morphology of the active layer after the reaction, the substrate was placed vertically in a stirred reactor. Atomic force microscopy (AFM) was employed to test the surface roughness. As shown in Figure 2a–c, the arithmetic mean surface roughness of PDA/ITO was *Ra* = 0.65 nm. The flatness was due to the self-assembly of PDA on the surface. After anchoring the initiator, the surface roughness increased marginally to 0.95 nm, but the overall surface roughness was still very low. After SI-ATRP, the overall roughness increased slightly, to 1.06 nm. Figure 2d shows the height profiles along three lines in Figure 2a–c. It can be clearly seen that the surface roughness gradually increased after the introduction of the initiator and iridium polymer. The thickness of the active layer was measured by field emission scanning electron microscopy (FESEM), as shown in Figure 2e,f. The thickness of the active layer after PDA self-assembly was approximately 53 nm, while the thickness increased to about 82 nm after SI-ATRP, indicating that the thickness of the grafted iridium polymer was 29 nm.

The Al/PDA-PPy_3_Ir/ITO device (Figure 3a) was prepared after evaporating aluminum as the top electrode on the active layer. The device exhibited typical nonvolatile rewritable resistive memory behavior in subsequent electrical performance tests, with turn-on and turn-off voltages of −1.0 V and 3.2 V, respectively, and a current ratio of 6.3 × 10^3^. When a negative scan voltage from 0 V to −3 V was applied, the current jumped from a low-conductive state (OFF) to a high-conductive state (ON) at −1.0 V. The corresponding current changed from 3.3 × 10^−6^ A to 2.1 × 10^−2^ A, which implied that the current switching ratio reached 6.3 × 10^3^. This process corresponded to the “write” process of data storage (sweep i). The device remained in a highly conductive state and could be read by another negative scan as the “read” process (sweep ii). Subsequently, a positive voltage was applied, and the conductivity of the device in the “ON” state dropped precipitously at the voltage threshold of 3.2 V (sweep iii).The conductivity remained low after applying another positive voltage sweep (sweep iv), which corresponded to the “erase” and “read” process of data storage, respectively. Similar cycles occurred in the subsequent repeated sweeps v–viii which demonstrated that the device possessed nonvolatile rewritable memory characteristics.

The effect between current and time was investigated by applying a constant voltage of −0.5 V. The current remained at about 10^−2^ A and 10^−6^ A, respectively, even on the time scale of 10^4^ s (Figure 3c). Large currents were very rare, indicating high stability of the device during continuous operation. Pulse voltage tests were also used to evaluate the stability of the device. More than 10^8^ times reading pulse tests were performed with a pulse period of 20 μs and width of 10 μs under an applied voltage of −0.5 V. As a result, the device was insensitive to pulse voltage stimulation since the current did not change significantly (Figure 3d). During 200 switching cycles, as shown in Figure 3e, the device exhibited strong cyclic stability in both the ON and OFF states. Figure 3f gave the cumulative probability plots of ON and OFF state at −0.5 V, which exhibited a small variation.

Flexibility is the crucial property for PDA-based devices in application of flexible and wearable resistive switching memory. *I*–*V* characteristic measurement was applied under both tensile and compressive strain (Figure 4a). As shown in Figure 4b, the device exhibits a bistable switching behavior under tensile strain, with turn-on voltage of −1.5 V and turn-off voltage of 4.2 V. Similarly, the device still exhibits typical nonvolatile rewritable memory performance under compressive strain, with turn-on and turn-off voltage of −1.5 V and 4.5 V, respectively. The cumulative probability diagrams of the turn-on and turn-off voltages of the device under different strains show stable and concentrated distribution (Figure 4d). In addition, when tensile strain and compressive strain are applied, the current values of the device in the ON state and OFF state almost remain unchanged, and the ON/OFF current ratio is greater than 10^4^. 

To compare the differences between SI-ATRP and traditional spin-coating processes, an additional contrast device was designed. First, AIBN was used as an initiator, and the iridium complex was polymerized into an iridium polymer by free-radical polymerization [36] (Appendix A). After the self-assembly of PDA on the ITO substrate, the iridium polymer was spin-coated on the surface of the PDA film. The top aluminum electrode was evaporated to obtain an Al/PPy_3_Ir/PDA/ITO device. XPS spectra are shown in Appendix A. Subsequently, a morphology analysis was carried out (Appendix A). AFM images showed that the roughness of the active layer obtained by spin-coating was 2.02 nm, which was much greater than that of the Al/PDA-PPy_3_Ir/ITO device. Due to the large molecular weight of the monomer, the iridium polymer was prone to agglomeration instead of being evenly dispersed on the surface. A bright bulge emerged in the 3D AFM and FESEM images. The surface of PDA-PPy_3_Ir/ITO was more uniform than that of PPy_3_Ir/PDA/ITO, indicating that the SI-ATRP method was more suitable for forming a smooth active layer film.

In subsequent electrical performance tests, the Al/PPy_3_Ir/PDA/ITO device obtained by spin-coating also showed nonvolatile memory performance with turn-on and turn-off voltages of −1.8 V and 4.4 V, respectively, and a current ratio of 13.3 (Appendix A). Specifically, when the negative voltage was scanned from 0 to −2.5 V, the current jumped from 4.5 × 10^−3^ A to 6.0 × 10^−2^ A when the voltage reached −1.8 V. The device changed from a low-conductive state to a high-conductive state. The device remained in the high-conductive state when the negative voltage was reapplied. After applying a positive scanning voltage, the device returned to a low-conductive state once the voltage reached 4.4 V. It retained its low conductivity under a subsequent positive voltage, showing nonvolatile rewritable resistive memory behavior. Then, the reliability of the memory device was studied by measuring its electrical performance under a constant voltage and pulse voltage. When the constant voltage of −0.5 V lasted for more than 10^4^ s, the OFF state and ON state currents were about 10^−3^ A and 10^−2^ A, respectively, but there were sometimes large fluctuations. As the current switching ratio was only 13.3, there was a greater risk of misreading. A similar problem occurred during the pulse voltage test. Different from the stable distribution of current values in the Al/PDA-PPy_3_Ir/ITO device in the two conductive states, the device obtained by spin-coating showed large fluctuations during the switching cycle test. The device obtained by SI-ATRP had a higher current switching ratio, which reduced the misreading risk, and better time retention, pulse voltage endurance, and switching cycle stability than the device prepared by the spin-coating method.

Iridium complexes have been previously used as electron acceptors in resistive memory devices [20,37], so the non-volatile rewritable characteristics of the Al/PDA-PPy_3_Ir/ITO device may be assigned to the charge transfer between iridium polymer and PDA. The existence of charge transfer was tested by spectroscopy. The UV-vis spectra of the active layer material PDA-PPy_3_Ir in different solvents are shown in Figure 5a. A wide absorption band appeared in the range of 325–425 nm. The characteristic peak appeared at 382 nm in toluene and shifted to 378 nm and 375 nm when the solvent was changed to the more polar tetrahydrofuran (THF) and *N*,*N*-dimethylformamide (DMF). This blue-shift indicated that the absorption may be attributed to the n-π* transition of the iridium complex [37]. A similar phenomenon also appeared in the steady-state fluorescence spectra in Figure 5b. In toluene, only one emission band at 512 nm appeared when the excitation wavelength was 390 nm. When THF and DMF were used, the peak position gradually shifted to 515 nm and 521 nm, respectively. The emission peak position exhibited a red-shift of 9 nm and its intensity decreased significantly, which indicated that charge transfer may exist in PDA-PPy_3_Ir.

To provide evidence of intramolecular charge transfer, electron paramagnetic resonance (EPR) tests of the active layer material before and after UV illumination were carried out. The result in Figure 5c showed that the active layer had a strong EPR signal with a *g* value of 2.0023 and a peak-to-peak width (Δ*H*_pp_) of 7 G before illumination. After UV irradiation for 5 min (wavelength 365 nm), the *g* value and Δ*H*_pp_ remained almost unchanged (*g* = 2.0024, Δ*H*_pp_ = 6 G). However, the signal intensity was obviously weakened, which may be attributed to charge transfer between PDA (electron donor) and iridium polymer (electron acceptor) after irradiation, while unpaired electrons or radicals recombined. The existence of light-induced intramolecular charge transfer was proved [38].
*E*_HOMO_/*E*_LUMO_ = −(*E*_ox/red_ − *E*_ox.(ferrocene)_) − 4.8 eV(1)
*E_g_* = *E*_LUMO_ − *E*_HOMO_(2)

The cyclic voltammetry of the active layer showed multiple oxidation and reduction peaks in Figure 5d. According to the equations above, where *E*_HOMO_, *E*_LUMO_, *E*_ox/red_, *E*_ox.(ferrocene)_, and *E_g_* represent the energy level of the highest occupied molecular orbital (HOMO), lowest unoccupied molecular orbital (LUMO), oxidation/reduction potential of the material, the oxidation potential of ferrocene (0.38 eV) in this electrochemical system, and the bandgap of the material, respectively [39]. Since the oxidation potential was 0.86 V, and the first reduction potential was −0.77 V, the potentials of the HOMO and LUMO were calculated to be −5.28 eV and −3.65 eV, relative to the saturated calomel electrode (SCE) after correction. For the electrode material, the potential of ultra-pure Al, as the top electrode, was −4.28 eV relative to the SCE electrode, while the potential of ITO, as the bottom electrode, was −4.8 eV. The energy barrier between ITO and the HOMO energy level was smaller than that between Al and the LUMO energy level (Figure 5e). This indicates that charge transfer in the active layer was dominated by hole transfer [39]. Under the applied electric field, efficient hole transfer from donor (PDA) to acceptor (iridium polymer) occurs in Al/PDA-PPy_3_Ir/ITO device. When the charge transfer state is reached, the conductivity of the D-A system rises sharply, which corresponds to the device switching to the high conduction state (ON) after the SET process (Appendix A). The AC impedance spectra (Figure 5f) of the active layer showed that the total resistance of the active layer was lower than that of pure PDA. The equivalent circuit of the entire system was R(CR)(CR)W, and the resistance was composed of the total resistance of the solution (Rt), contact resistance (Rs), and charge transfer resistance (Rct). Since the total resistance of the solution was the same as that of the electrolyte solution, and the contact resistance between the platinum electrode and the solution was identical, the difference of resistance was attributed to the lower charge transfer resistance of PDA-PPy_3_Ir [40]. 

For the Al/PPy_3_Ir/PDA/ITO device obtained by the spin-coating method, a donor-acceptor heterojunction memory device with a structure of Al/acceptor polymer/donor polymer/ITO was constructed. The iridium polymer was the electron acceptor layer, and PDA was the electron donor layer. This spin-coating method relied on the physical connection between layers and was weaker than covalent bonds between iridium polymer and PDA. This may be the reason why the device obtained by the SI-ATRP method was more reliable than the one obtained by the spin-coating method.

## 4. Conclusions

Here, PDA was self-assembled on the surface of an ITO/PET substrate through bio-inspired mussel chemistry, and then an iridium polymer was grafted from the ITO/PET substrate via PDA-mediated SI-ATRP to fabricate the Al/PDA-PPy_3_Ir/ITO flexible device. The device exhibited nonvolatile rewritable memory characteristics with a turn-on voltage of −1.0 V and an ON/OFF current ratio of 6.3 × 10^3^. Furthermore, the memory behaviors of the flexible device were almost unchanged under the bending test. Compared with the contrasting Al/PPy_3_Ir/PDA/ITO device obtained by the spin-coating method, the Al/PDA-PPy_3_Ir/ITO device obtained by SI-ATRP reaction had lower risk of misreading and higher reliability. The nonvolatile memory characteristics of the device were derived from the charge transfer in the active layer material. This work provides a simple and novel strategy for the fabrication of flexible organic resistive memory devices.

## Data Availability

Not applicable.

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
