# Peer review of "In-Situ Surface Modification of ITO Substrate via Bio-Inspired Mussel Chemistry for Organic Memory Devices"

_biomimetics, 2022, doi:10.3390/biomimetics7040237_

Round 1

Reviewer 1 Report

This manuscript “In-situ surface modification of ITO substrate via Bio-inspired mussel chemistry for organic memory devices” by Minglei Gong et al. demonstrated the two terminal memory devices consisting of Al/PDA-PPy3Ir/ITO. A device’s performance is further compared in the case of film preparation as spin-coating and SI-ATRP techniques. The manuscript still lacks the deeply discussion regarding the device working mechanism and there are a lot of mistakes during the writing. One of the important of this work is taking an advantage of flexible substrates, however, there is not any result related to this. I recommend the author to carefully revise the manuscript before considering publishing this study.

1.     The author should provide the memory behavior based on the flexibility.  

2.     Regarding the schematic illustration of the Al/PDA-PPy3Ir/ITO device, it is necessary to show the flexible PET substrate as well.

3.     The abstract was written very poor, it needs to revise carefully. It is also important to show the comparison of your work to previous studies. What is exactly “better time retention”, and “stronger pulse voltage endurance”, and “switching cycle stability”?

4.     In the page 2 (line 70-72), it is not correct that the SI-ATRP had a higher reliability, including retention time, pulse voltage tolerance, and switching cycle stability. As results shown in Figure S5 (supporting information), the spin-coat device still showed a same aspect as SI-ATRP device; there is only different in case of on/off current ratio. What is a main reason to have a huge difference in the Off-current of the spin-coat device compared to the SI-ARRP one?

5.     What is blue and red curve as shown in Figure S5? Is there any discussion regarding the on/off current is lower than SI-ATRP device?

6.     The y-axis in Figure 1b, c, d is missing.

7.     The scale-bar in the AFM images (Figure 2a-c) is missing.

8.     I recommend using the label of sweeping voltage direction i-viii rather than 1-8 (Figure 3b). And the step 2,6 is wrong. How much voltage did you apply for SET/RESET process as Figure 3c-e?

9.     Is there any reason to sweep the voltage from the negative side to positive side? What if the device is swept from positive to negative side?

10.  It is not clear about the device working mechanism, I suggest drawing the band diagram to discuss in detail.

11.  It is important to show the cumulative probability a long with the resistance of memory device. The author should either show this plot or mentioned it in case of device variation in the main manuscript.

Reviewer 2 Report

In this manuscript, the authors report a novel device preparation method that uses mussel-inspired chemistry to self-assemble the polydopamine (PDA) coating on the substrate, followed by in situ growth of functional units on the PDA surface by surface-initiated atom transfer radical polymerization (SI-ATRP), and finally deposit the top electrode to form a sandwich-structured device. Compared with the traditional spin-coating method, this PDA-mediated SI-ATRP method has obvious advantages, and the prepared device has a larger ON/OFF current ratio of 6.3×103, better time retention, stronger pulse voltage endurance, and switching cycle stability. This work not only provides a reference for the research of organic memory devices, but also the method adopted can be extended to other research fields. Therefore, I will recommend it for publication in Biomimetics after minor revisions noted below.

Q1.
The authors only tested the performance of the device in the normal state, so it is not reasonable to draw the device schematic (Figure 3a) in the bent state, and the authors should correct it.

Q2. There are some formatting problems in the manuscript, for example PDA-PPy3Ir in the caption of Figure 1-4 should be corrected to PDA-PPy3Ir.

Q3. The authors need to explain why the device performance of Al/PDA-PPy3Ir/ITO is better than Al/PPy3Ir/PDA/ITO.

Q4 Some outstanding research advances in related fields should be incorporated, few are given below.

Adv. Funct. Mater. 2022, 32(8): 2108598. Adv. Funct. Mater. 2021, 31(21): 2100144. Adv. Funct. Mater. 2019, 29(3): 1806637. Adv. Electron Mater. 2019, 5(4): 1800793.

Round 2

Reviewer 1 Report

The revised version is significantly improving the quality of manuscript, all questions are clearly clarified. Therefore, I suggest accepting this study to publish in the Biomimetics.

Minor: Please check the caption of Figure 3f, “weibull”?

Author Response

Thank you for recommending our manuscript to be published in Biomimetics.

Q1. Please check the caption of Figure 3f, “weibull”?

Reply: Many thanks. Fig. 3f shows the Weibull distribution of the device resistances in the ON and OFF states. We have changed the caption of Fig. 3f in our revised manuscript. " (f) Cumulative probability plots of the ON and OFF states. The resistances are acquired at −0.5 V."